# Progression of Myeloproliferative Neoplasms (MPN): Diagnostic and Therapeutic Perspectives

**DOI:** 10.3390/cells10123551

**Published:** 2021-12-16

**Authors:** Julian Baumeister, Nicolas Chatain, Alexandros Marios Sofias, Twan Lammers, Steffen Koschmieder

**Affiliations:** 1Department of Hematology, Oncology, Hemostaseology and Stem Cell Transplantation, Faculty of Medicine, RWTH Aachen University, 52074 Aachen, Germany; jbaumeister@ukaachen.de (J.B.); nchatain@ukaachen.de (N.C.); 2Center for Integrated Oncology Aachen Bonn Cologne Düsseldorf (CIO ABCD), 52074 Aachen, Germany; asofias@ukaachen.de (A.M.S.); tlammers@ukaachen.de (T.L.); 3Department of Nanomedicine and Theranostics, Institute for Experimental Molecular Imaging, Faculty of Medicine, RWTH Aachen University, 52074 Aachen, Germany

**Keywords:** MPN, progression, diagnosis, therapy

## Abstract

Classical BCR-ABL-negative myeloproliferative neoplasms (MPN) are a heterogeneous group of hematologic malignancies, including essential thrombocythemia (ET), polycythemia vera (PV), and primary myelofibrosis (PMF), as well as post-PV-MF and post-ET-MF. Progression to more symptomatic disease, such as overt MF or acute leukemia, represents one of the major causes of morbidity and mortality. There are clinically evident but also subclinical types of MPN progression. Clinically evident progression includes evolution from ET to PV, ET to post-ET-MF, PV to post-PV-MF, or pre-PMF to overt PMF, and transformation of any of these subtypes to myelodysplastic neoplasms or acute leukemia. Thrombosis, major hemorrhage, severe infections, or increasing symptom burden (e.g., pruritus, night sweats) may herald progression. Subclinical types of progression may include increases in the extent of bone marrow fibrosis, increases of driver gene mutational allele burden, and clonal evolution. The underlying causes of MPN progression are diverse and can be attributed to genetic alterations and chronic inflammation. Particularly, bystander mutations in genes encoding epigenetic regulators or splicing factors were associated with progression. Finally, comorbidities such as systemic inflammation, cardiovascular diseases, and organ fibrosis may augment the risk of progression. The aim of this review was to discuss types and mechanisms of MPN progression and how their knowledge might improve risk stratification and therapeutic intervention. In view of these aspects, we discuss the potential benefits of early diagnosis using molecular and functional imaging and exploitable therapeutic strategies that may prevent progression, but also highlight current challenges and methodological pitfalls.

## 1. Introduction

Classical Philadelphia chromosome-negative (Ph^-^) myeloproliferative neoplasms (MPN) are a group of clonal myeloid stem cell disorders comprising essential thrombocythemia (ET), polycythemia vera (PV), and primary myelofibrosis (PMF). A common feature, comprising over 90% of MPN cases, is the presence of a driver mutation in the genes encoding janus kinase 2 (*JAK2*), calreticulin (*CALR*), or the thrombopoietin receptor (TPOR, *MPL*) [1,2,3]. These oncoproteins induce constitutive activation of the JAK2 signaling pathway, leading to hyperproliferation of mature myeloid cells. Moreover, the progressive loss of heterozygosity and ensuing enhanced mutational allele burden during disease development (e.g., by uniparental disomy or chromosomal duplication) may result in increased cytokine-independent proliferation of the hematopoietic clone and provide a competitive advantage over its normal counterparts.

Mechanistic features of MPN progression comprise genetic factors such as enhanced mutational allele frequencies of oncogenic drivers, as well as increased cytokine-independent proliferation of the hematopoietic clone. Moreover, the acquisition of additional mutations in genes encoding for epigenetic regulators, transcriptional regulators, splicing factors, or (other) tumor suppressors may enhance the fitness and proliferative advantage of the malignant clones and it may also make them less responsive to anti-MPN therapies [4]. Loss of function mutations (and in some cases gain of function mutations) contribute to progression to a more myelodysplastic and/or myelofibrotic phenotype, leading to an accelerated phase and eventually a terminal blast phase of the MPN, which typically presents as an acute myeloid leukemia (AML) with poor response to therapy. As a result of the recent advances in sequencing technologies, the implication of these genetic factors for MPN progression has gained much attention.

Besides these genetic factors, further factors have been demonstrated to deteriorate the disease course. They include cytokine-driven inflammation by the malignant clone, but also by non-malignant hematopoietic cells and by non-hematopoietic cells of the bone marrow (BM) microenvironment. In particular, the latter contribute to inflammation by (a) the excess production of fibrotic tissue through activation of so-called myofibroblasts which leads to suppression of normal hematopoiesis, (b) enhanced cytokine production that favors the malignant clone (e.g., TNF-alpha), and (c) abnormal expression of thrombogenic surface molecules on endothelial cells which increase the risk of thrombosis (e.g., P-selectin) [5,6,7,8]. Finally, systemic conditions such as chronic inflammation, cardiovascular diseases, and pro-fibrotic processes in other organs (kidney fibrosis) may augment the risk of MPN progression [9].

Collectively, these risk factors for MPN progression and their dynamic changes over time will determine the disease course of an individual patient. Clinically, this may be evident by the development of thrombosis or thromboembolism, major bleeding, severe infections which poorly respond to antibiotic therapy, or increasing symptom burden, such as painful splenomegaly, unintended weight loss, night sweats, fever, and pruritus. These symptoms may indicate progression of a more indolent disease (ET, PV, and pre-PMF) to an aggressive disease (overt PMF, pPV-MF, pET-MF, MPN-AP, or AML). In addition to these clinically evident types of MPN progression, the detection of subclinical MPN progression will be essential for optimal counseling and management of MPN patients. Thus, indicators of subclinical progression, such as biomarkers for bone marrow fibrosis, driver oncogene mutational allele frequency, or quantifiers of clonal evolution, may become increasingly important in the near future.

Current therapeutic approaches focus both on the prevention (e.g., primary prevention of thrombotic events) and treatment of MPN progression (e.g., delay of overt MF development). These approaches have already led to significant improvements in symptom control and cardiovascular morbidity, and some of these treatments may prolong survival, including JAK inhibitors, interferons, and potentially curative allogeneic stem cell transplantation (aSCT).

In this review, we address the types and mechanisms of MPN progression and how they are currently quantified as well as ongoing novel therapeutic approaches and their potential to modify the disease course.

## 2. Types of MPN Progression

### 2.1. Clinically Evident Types of MPN Progression

Clinical indicators of MPN progression include the development or aggravation of MF and perturbance of blood counts, such as elevated white blood cells (WBC), the presence of granulocytic precursor cells and blasts, or reduced red blood cells. Although MPN are considered chronic diseases, they have the potential to progress to acute leukemia, also termed blast phase MPN (MPN-BP), which is characterized by >20% blasts in peripheral blood (PB) or BM (Figure 1). An AML that originates from an antecedent MPN is different from de novo AML, presenting with unique molecular and cytogenetic features [10]. In contrast to chronic phase MPN (MPN-CP), MPN-BP patients have a severely reduced survival (median survival of 3.6 months), and, to date, no therapeutic modality has consistently led to prolonged survival [11,12], even though a small fraction of patients can be rescued by chemotherapy and subsequent allogeneic transplantation. In analogy to BCR-ABL-positive CML, patients may present in an intermediate phase between MPN-CP and MPN-BP, defined as accelerated phase (MPN-AP), which is characterized by 10–19% circulating blasts, although some prognostic scoring systems, which are discussed in more detail in Section 4, suggest even lower blast counts as risk factors in Ph-MPN [13,14].

Despite transformation into AML, some of the different MPN subtypes have the potential to progress into each other (Figure 1). While ET patients can progress into PV [15], both carry the risk of progressing into post-ET-MF or post-PV-MF (also termed secondary MF (sMF)), occurring in approximately 15% of patients and also the risk of leukemic progression. Prefibrotic PMF (Pre-PMF), a disease state that mimics ET but shows additional abnormalities in the granulocytic lineage, has a risk of progressing to overt PMF or AML in 15.2% or 4.7%, respectively [16,17]. Other less common types of disease progression include absolute monocytosis in PV and PMF patients or neutrophilic leukocytosis in pPV-MF, both of which are associated with a shorter overall survival (OS) and accelerated progression [18,19].

Symptomatically, the occurrence or intensification of thromboembolic events, major bleedings, and constitutional symptoms such as pruritus, night sweats, fever, weight loss, or fatigue is associated with progression. In addition to these clinically defined forms of MPN progression, progression may be detectable at the cellular or molecular level. Karyotypic changes are considered one of the most important genetic predictors of leukemic progression. While chromosomal abnormalities are relatively rare events in ET and PV, they were identified in 46% of PMF [20]. As the acquisition of chromosomal aberrations is a common feature during accelerated and blast phases [21], this might explain the relatively high risk of transformation in PMF (10–20% at 10 years), compared with ET and PV (1–2.3% at 10 years) [12,22,23].

While it is not completely understood how one mutation, *JAK2*V617F, can lead to three different disease phenotypes, the presence of a higher allelic burden is associated with a PV [24] or pPV-MF phenotype. In general, *JAK2*V617F AB positively correlated with hemoglobin concentration, WBC count, spleen size, and BM cellularity, but also with an increased risk of myelofibrotic progression [25]. In contrast, no association was found between *JAK2*V617F allelic burden and the risk of leukemic progression. Instead, additional factors, such as mutations of *TP53* and *IDH1/2*, were found to correlate with progression to MF and/or AML [26] and are primarily found in overt PMF and MPN-BP, but rarely in ET or PV [27,28,29]. Somatic bystander mutations associated with leukemic progression that are also commonly mutated in patients with long-standing MPN-CP include epigenetic regulators, transcription factors, and spliceosomal genes and are discussed in Section 3.1. These mutations inversely correlated with OS and were incorporated into various genetically based scoring systems [14,30,31,32]. Finally, clinical progression of an MPN could also be quantified using the MPN-specific prognostic scoring systems, where advancement to a higher risk group in such a prognostic scoring system (further discussed in Section 4) could be considered an indicator of disease progression.

### 2.2. Subclinical Types of MPN Progression

The allelic burden of driver mutations is a highly variable factor in MPN. A large highly sensitive droplet digital PCR study including 19,958 participants revealed the presence of *JAK2*V617F and *CALR* mutations in 3.1% and 0.16%, respectively, of individuals in the general population, suggesting a much higher prevalence than previously expected [33]. Although for most participants, the allelic burden was well below the detection limit of other methods such as Next Generation Sequencing (NGS) and digital droplet-PCR (ddPCR), a distinct subclinical profile mimicking an MPN phenotype with elevated blood cell counts was observed for individuals with an allelic burden around 1%.

These observations suggest that individuals harboring the *JAK2*V617F mutation with very low allelic burden are at risk to develop MPN, although it is uncertain in how many individuals progression to an overt MPN might eventually occur throughout their life. Williams et al. reconstructed the phylogeny of hematopoiesis by tracing 448,553 somatic mutations by whole-genome sequencing and discovered that the *JAK2*V617F mutation was acquired in early childhood or even in utero, and it is currently unclear which factors determine the long latency from acquisition to clinical presentation, the mean of which was 31 years [34]. This finding suggests that the low clonal penetrance of *JAK2*V617F might require other (epi-) genetic alterations.

The phylogenetic reconstructions suggest that the time of MPN diagnosis is merely one time-point within a lifelong trajectory of clonal evolution, at which symptom burden or blood cell counts have surpassed a certain level, or at which clinical complications have occurred [34]. This knowledge, together with new sequencing technologies, opens opportunities for early detection of an emerging malignant clone and intervention before it manifests itself as an overt MPN.

In addition to increasing allele burden of driver mutations and the emergence of new additional mutations in subclones, termed clonal evolution, subclinical disease progression can also be detectable as an increase in BM fibrosis grade. This is the case when patients with ET or PV progress to “ET/PV with BM fibrosis” without meeting the full IWG-MRT criteria for sMF (Figure 1) [35]. However, it is currently unknown whether development of such BM fibrosis is associated with an inferior survival.

## 3. Etiology of MPN Progression

### 3.1. Genetic Risk Factors for MPN Progression

During the past decade, next generation sequencing and other sequencing-based methods have significantly enhanced the identification of genetic factors that drive clonal dominance and potentially promote progression. Driver mutations alone might play a subordinate role in disease progression, as most PV patients with a very high allelic burden remain in chronic phase for a long time, while triple-negative MF patients (harboring no *JAK2*V617F, *CALR* or *MPL* mutation) have an elevated risk of leukemic transformation [36]. In contrast, a variety of mutations in epigenetic modifiers (*TET2*, *DNMT3A*, *IDH1/2*), transcription factors (*TP53*, *RUNX1*, *IKZF1*), and splicing factors (*SF3B1*, *U2AF1*, *SRSF2*) have been identified and assigned a role in progression from ET or PV to sMF or from MPN-CP to MPN-BP, with an associated myelodysplastic phenotype that increases with the number of these mutations [4]. This is supported by the fact that these mutations are even more frequently mutated in myelodysplastic syndromes (MDS) and AML [37]. Some of these mutations modify stem and progenitor cell function and have an intricate role in clonal hematopoiesis of indeterminate potential (CHIP), resulting in skewing towards myeloid differentiation. Mutations of *ASXL1*, *IDH1/2*, *EHZ2* or *SRSF2* are detected in every third PMF patient and are associated with shorter OS and leukemia-free survival [38]. With regards to fibrotic progression, no association with the CHIP-associated mutations *TET2*, *ASXL1*, *DNMT3A* was found; in contrast, mutations of *SRSF2*, *U2AF1*, *SF3B1*, *IDH1/2*, and *EZH2* that are rarely found in CHIP showed a strong correlation with fibrotic progression [39].

At a larger scale, chromosomal aberrations are one of the predominant features of MPN progression. While such chromosomal aberrations are a relatively rare event in Ph-negative MPN-CP, their occurrence is wide-spread in post-MPN AML patients [40]. Those patients harboring chromosomal aberrations showed features of disease progression and frequently developed AML at a later follow-up.

### 3.2. Inflammation

Apart from the aforementioned genetic causes, inflammatory processes were not only assigned a role in triggering symptoms in MPN patients, but also in progression to overt MF or BP-MPN. The extensive interaction between the malignant clone, non-malignant hematopoietic and non-hematopoietic cells in the BM microenvironment (including mesenchymal stromal cells [MSCs]), and inflammatory cytokines or reactive oxygen species (ROS) is a powerful driving force of fibrotic remodeling (as reviewed in [41]). In this process, dysplastic megakaryocytes are of particular importance: they secrete a plethora of inflammatory cytokines which evoke the reprogramming of endothelial cells or MSCs into myofibroblasts, and these myofibroblasts are key factors in fibrotic tissue formation. Increased levels of interleukin 8 (IL-8/CXCL8) and IL-1β were identified as prognostic indicators for the progression of PMF to MPN-BP, and TNF-alpha correlated with MPN-BP progression irrespective of the MPN subtype [5,42,43]. Intriguingly, elevated IL2R and IL-8 plasma levels alone were correlated with inferior survival in PMF patients [5]. Other proinflammatory cytokines that are elevated in MPN include, but are not limited to, IL-6, IL-8, IL-15 VEGF, b-FGF, TGF-β, HGF, EGF, and GRO-α [44,45]. The dysregulation of pro-inflammatory cytokines is independent of the driver mutation, as also non-malignant cells in MPN patients show elevated levels and was identified an important driving force for clonal evolution and disease progression [46,47]. Moreover, several studies reported elevated levels of reactive oxygen species (ROS) in MPN, and these molecules are key signaling molecules in the progression of inflammatory disorders [48,49]. Furthermore, they facilitate clonal dominance, by enhancing JAK-STAT signaling, and they induce oxidative DNA damage, causing genomic instability and thereby promoting disease progression to overt MF or MPN-BP through the acquisition of additional mutations [49,50].

Hence, suppression of chronic inflammation in MPN by anti-inflammatory drugs is a promising strategy to alleviate disease-associated symptoms and to escape the vicious cycle that promotes progression. While some of the aforementioned cytokines are insensitive to JAK inhibition, IFNa may exert different actions by the repression of IL-1β, IL-11, HGF, and TGF-β [43]. This JAK2-independent mechanism may thus contribute to the efficacy of (ro-)peginterferon. In addition, direct blocking of IL-1 β may be of therapeutic use in MPN [51,52]. Several clinical studies are ongoing that investigate the potential of anti-inflammatory and immunomodulatory drugs (see Section 6). Although substantial progress has been achieved regarding the role of inflammation in MPN pathophysiology, several questions remain unanswered, including the role of immune escape or immunogenic cell death, as discussed in Section 6.

### 3.3. Age and Comorbidities

MPN typically become clinically apparent in elderly patients, with the mean age at diagnosis of ET, PV, PMF, pPV-MF, and PET-MF being 52.9, 59.2, 60.6, 58.4, and 63.9 years, respectively [53]. Younger PV patients were generally found to harbor a single somatic JAK2V617F mutation, whereas older patients had higher frequencies of bystander mutations [54]. In line with this finding, older patients had a shorter median disease duration until progression to sMF or MPN-BP, although the frequencies were similar [55].

Age-associated inflammation was linked to the upregulation of NF-κB signaling, modulated cytokine secretion, telomere shortening, ER stress, and lipid accumulation with increasing age [56]. Therefore, the inflammatory milieu drives clonal expansion predominantly in the elderly population.

Thus, age can be considered a third factor for disease progression and has been incorporated into various prognostic risk scores.

Comorbidities can affect the rate of thrombotic and hemorrhagic complications in MPN patients. For example, kidney dysfunction has been associated with an increased risk of thrombosis in MPN [9,57,58]. However, whether comorbidities also affect the rate of progression in MPN patients is largely unknown. Interestingly, one study found that a higher body mass index (BMI ≥ 25) was associated with a lower probability of progression to pPV-MF and improved survival [59]. The authors speculate that normal weight or underweight may be due to a hypercatabolic state that can be associated with disease progression [60].

## 4. Clinical Risk Scores

When caring for patients with MPN, one major challenge is to untangle the pathogenic complexity and stratify patients into clinically actionable subsets, estimating their survival and risk of fibrotic progression and leukemic transformation, and recommending personalized treatment. Over the past decade, several prognostic scoring systems have been established (Table 1). The earliest studies were limited by the rarity of the disease subtypes and insufficient understanding of disease pathophysiology, resulting in the early conventional prognostic models that were restricted to patient age and history of thromboembolic events [61]. Over time, larger patient cohorts enabled the incorporation of clinical parameters such as blood cell counts (leukocytes, blasts, platelets, and erythrocytes) and constitutional symptoms to estimate survival and risk of thrombosis or leukemic transformation in MPN patients. The identification of *JAK2*V617F, *CALR*, and *MPL* mutations as molecular drivers of the diseases paved the way to more precise diagnostics, but also significantly improved prognosis after implementation in the mutation-enhanced scoring system MIPSS70 for PMF [14]. Other genetic parameters that were included in these scoring systems over time are cytogenetic abnormalities and recurrent prognostically detrimental non-driver mutations that are described in Section 3.1.

While scoring systems for ET and PV implemented OS and the thrombotic risk, PMF scoring systems prognosticate OS and the risk of leukemic transformation. Some of these risk scores were developed for specific purposes, with MYSEC for sMF patients (that biologically and clinically differ from PMF patients) or MTSS for patients undergoing transplantations [69,70]. A recent tool for personalized risk calculation also allows the prediction of fibrotic progression and transformation into AML in several MPN subtypes [31]. This tool is built upon comprehensive genetic information and may be helpful in selecting patients with a very high risk of progression to MF and/or AML.

## 5. Current MPN Diagnostic Landscape and New Directions

The early detection of disease onset or progression into MF or MPN-BP is one of the key challenges in the management of MPN patients.

The low clonal penetrance of *JAK2*V617F, as indicated by the observation that typically several decades pass before the *JAK2*V617F-positive clone manifests clinically, provides the opportunity for achieving early detection and intervention before clonal hematopoiesis gives rise to an MPN [33,34]. In addition, the importance for early detection and intervention is underlined by the fact that thrombohemorrhagic complications can precede MPN diagnosis by several years [71,72]. Based on the identification of MPN patients exhibiting BM changes, but normal cell counts, LDH, and spleen size, the existence of a disease stage reflecting *carcinoma-in-situ* in solid tumors was hypothesized [73]. At this early stage, the localization of the malignant clone was suggested to be restricted to primary clonal proliferative spots within the BM. If this hypothesis, which is supported by the heterogenic spatial disease distribution observed in AML [74], is proven true, diagnosing MPN and AML at early disease stages by genomic analyses would be insufficient, since (a) blood sampling may not contain any traits of the malignant clone, and (b) BM sampling may fail to collect and consequently detect malignant clone hidden “hot-spots”.

Nevertheless, as of now, BM biopsies are a standard procedure in diagnosis and assessment of treatment response in MPN patients. This invasive technique, although rarely associated with complications, bears certain risks such as bleeding and infection. BM biopsy is also hampered by methodological pitfalls, such as overlapping features between ET, PV and PMF, sampling errors due to non-homogenous disease distribution throughout the body, limited serial monitoring, and sparse information on functional processes [75]. Over the past decades, ultrasound (US), computed tomography (CT), positron emission tomography (PET), and magnetic resonance imaging (MRI) have emerged as important non-invasive diagnostic modalities in oncology. Although they convey a plethora of benefits (e.g., simplicity and low-cost (US), high spatial resolution for both soft and dense tissue (MRI, CT), high sensitivity (PET), molecular specificity (molecular PET)) that can substantially improve diagnostic efforts and longitudinal monitoring of cancer patients, they may be limited by radiation exposure (PET, CT), tissue penetration (US), and relatively low contrast-agent-based sensitivities (MRI, CT). Thus, combination of these techniques including structural, functional, and molecular imaging [76] may display the potential to overcome early detection sensitivity issues and differentiate among the various MPN stages.

A recent review summarized the implication of a variety of imaging techniques in MPN and concluded that MRI is a suitable tool for the evaluation of BM fat content as an indirect correlate of BM cellularity in PMF and potentially ET and PV [75]. Studies utilizing molecular imaging with ^18^F-fluoro-deoxyglucose (FDG), and ^18^F-fluoro-thymidine (FLT) PET-CT demonstrated the potential to differentiate early from late MF phases, based on the assumption that cellular ^18^FDG uptake or ^18^FLT incorporation into DNA are surrogate markers of inflammation or proliferation magnitude, respectively [77,78]. ^18^FLT-PET-CT might also be able to predict leukemic evolution, characterized by a progressive increase in BM activity [79]. However, due to the initial hyper-inflammatory/-proliferative phase in the BM of MPN patients, discrimination between early MF stages turned out to be challenging [77,78]. To directly assess myelofibrosis, we suggest that (a) low-dose quantitative CT [80] or (b) fibrosis-specific functional or molecular imaging [81,82] might aid in detection of progression from ET/PV to sMF and discrimination between the various early stages. In this regard, future studies should address the identification of the primary clonal proliferative spots in the BM that typically remain hidden from the blood or BM biopsies.

## 6. Novel Therapeutics Targeting MPN Progression

Cytoreductive drugs like hydroxyurea and anagrelide are efficient in preventing vascular events in MPN patients, which are the main objectives in ET and PV. JAK2 inhibitors such as ruxolitinib have shown potential to reduce splenomegaly and other disease-related symptoms and potentially improve survival of MF patients. However, to the best of our knowledge, these drugs have minor disease-modifying potential, at least when used as monotherapy, and are not capable of eradicating the malignant clone [83]. Although substantial progress has been made in improving diagnosis and prediction of disease course, the therapeutic options to prevent progression from CHIP to MPN, ET/PV to sMF, and MPN-CP to MPN-BP are still very limited. To date, alloSCT is the only therapy with curative potential; however, it can only be performed in a small subset of patients, due to the risks of transplantation-related morbidity and mortality. Two key challenges in MPN research are the identification of novel therapeutics that can modulate the disease course and, given the low overall survival of MPN-BP and its resistance to therapy, to prevent leukemic progression. Several clinical trials are ongoing that assess the potential of novel drugs exploiting different mechanistic concepts to mitigate the risk of MPN progression [84].

Interferon-alpha (IFNa) has the potential to achieve partial or, in some patients, even deep molecular remission, particularly in patients harboring the *JAK2*V617F mutation [85,86,87]. Although the underlying mechanism has not yet been fully elucidated, IFNa is known to directly target malignant HSCs, leading to increased STAT1 phosphorylation, elevated ROS, DNA damage, and reduced quiescence [88]. Since the introduction of pegylated (PEG)-IFNa, which reduces both dosing intervals and side-effects compared with the non-pegylated form, IFNa has regained interest as a potential tool to interfere with the disease course, via potently inducing hematologic and molecular responses in a majority of ET and PV patients [87]. In MF patients, the overall response rate (ORR) is lower than in ET or PV patients, and complete responses are rare [89]. In this subtype, combination treatments with ruxolitinib may achieve better results [90,91]. As chronic inflammation is a driving force of disease progression, this combination treatment or even triple-therapy with hypomethylating agents, such as LSD1 inhibitors that downregulate the expression of diverse inflammatory cytokines upregulated in MPN, might show efficacy in preventing disease progression [92,93].

As inflammation is an integral characteristic of MPN pathophysiology and closely involved in disease progression, the efficacy of novel immunomodulatory drugs such as BET, BCL2, MDM2, and telomerase inhibitors as well as TGF-β superfamily ligands and interferons is currently being tested in a variety of clinical studies (Figure 2, Table 2). One example is the bromodomain and extra-terminal domain (BET) inhibitor-mediated targeting of inflammatory NF-κB signaling, a pathway which is implicated in the production of pro-inflammatory cytokines and fibrosis. NF-κB signaling is activated in all MPN subtypes, but particularly sMF [94]. An interim analysis of several trials, including the global MANIFEST-2 trial, suggested improvements of BM fibrosis and a potential for disease modification by BET inhibition [95]. In addition, optimization of BET small molecule inhibitors by linkage to so-called PROTACs, specifically targeting the bound protein to proteolysis, determine a novel class of BET inhibitors [96]. Ultimately, in a personalized medicine approach, knowledge of the impact of progression-associated cytokines could lead to a more specific targeting of cytokines and thus represent an additional option for patient-specific symptom control and progression prevention. Therapeutic cancer vaccination against mutated CALR was suggested as a new treatment modality in CALR-mutated MPN. Although many patients displayed T cell responses, unfortunately, no clinical responses have been observed so far [97].

Most immunotherapeutic approaches were unable to show strong efficacy, which might be related to tumor immune escape evoked by the inflammatory microenvironment [98]. PD-L1, which is highly expressed in several cancers and in MPN [94,99], has a role in facilitating immune escape by suppressing T cell responses upon binding to its receptor PD-1. Although the initial clinical trials with the PD-L1-inhibiting monoclonal antibodies nivolumab (NCT02421354) and durvalumab (NCT02871323) were terminated due to a lack of efficacy, other molecules are investigated in ongoing studies, including the HSP90 inhibitor PU-H71 in MF (NCT03935555) which may improve immunotherapy by upregulating interferon response genes [100,101]. CALR exposure on the surface of dying cancer cells induces phagocytic uptake by dendritic cells and is a hallmark of immunogenic cell death (ICD). In contrast, loss of the ER retention signal KDEL in CALR frameshift mutant protein leading to excessive secretion was demonstrated to exhibit immune-modulatory effects by suppressing antineoplastic immune responses mediated by phagocytes [102]. The resulting inhibition of ICD was hypothesized to negatively affect the efficacy of chemotherapeutic agents as well as PD-1 blockade. This immunosuppressive mechanism highlights immune-modulatory functions of mutated CALR exceeding oncogene signaling-dependent consequences. These findings provide a rationale to target immune escape in CALR mutated MPN, possibly by scavenging soluble CALR protein by monoclonal antibodies.

Besides the identification of new therapeutic pathways and targets, improving the performance of currently available agents through nanomedicine is of interest. Examples of clinically approved and already extensively used nanomedicines are pegylated IFNa in MPN and liposomal cytarabine/daunorubicin (co-entrapped in a fixed 5:1 ratio) in AML [103]. Drug encapsulation in nanomedicine formulations prevents renal clearance and enhances blood circulation. This promotes drug accumulation at pathological site and reduces high dosage-associated toxicity. Active targeting using fibrosis- or megakaryocyte-specific ligands might further enhance accumulation at fibrotic hotspots and/or cell-type-specifically target the malignant clone. Co-delivery of two anticancer agents enables simultaneous synergistic effects within the same pathological cell. The application of nanoparticles loaded with drugs and imaging probes provides a tool to simultaneously and non-invasively monitor disease progression, drug accumulation, and treatment responses. This theranostic approach might be relevant in the management of MPN patients and should be investigated.

The early detection of disease onset may enable new paths to prevent MPN initiation. Although the concept of surveilling or selectively destroying the CHIP clone before it can give rise to an overt disease was proposed several years ago [104], it has regained a lot of interest by the findings that CHIP/MPN-associated mutations were already acquired in early childhood or even in utero and thromboembolic events occur years before MPN diagnosis [34,73,105]. Therapeutic approaches that are capable to interfere with clonal evolution may eradicate the CHIP clone or hold it at bay, thereby preventing these complications and progression into an overt MPN. This strategy should be carefully evaluated for the benefits in reducing disease-associated risks and treatment-associated complications and might be of particular interest for individuals with increased thrombotic risk or other comorbidities.

## 7. Conclusions and Outlook

Although substantial progress in understanding MPN pathophysiology has been made in the past two decades, the variety of progression types as well as their underlying triggers remain largely obscure, hampering prediction and prevention of progression.

To date, therapeutic options are capable of reducing symptom burden (and partially even induce molecular remission), but not of effectively modulating the disease course in the majority of patients. The only treatment of curative potential remains alloSCT, which can only be applied in a minority of patients [106].

New diagnostic and therapeutic tools are urgently needed to optimize the treatment of MPN patients with a particular focus on disease progression, as overt MF and especially MPN-BP are associated with poor survival. NGS and other whole-genome analyses are excellent tools for the identification of novel somatic and germline mutations. Due to the decreasing costs and their increasingly widespread utilization in clinics, these technologies will contribute to the identification of novel (intergenic) mutations involved in MPN pathophysiology, including convincing associations of these mutations with disease progression. These techniques may be supplemented by alternative approaches to identify genes that contribute to MPN-BP progression such as screens involving CRISPR/Cas9 tumor suppressor gene libraries [107].

Functional and molecular imaging tools may also become increasingly important for the non-invasive and early detection of spatially heterogeneous disease and progression to overt MF and MPN-BP. Therapeutic and diagnostic approaches employing nanoparticles could enable the combination of functional or molecular imaging with targeted drug-delivery to visualize drug accumulation and treatment responses. A better understanding of the pathophysiology of MPN progression will improve the prediction of progression in individual patients and may lead to improved prevention strategies in patients at high risk of progression.

## Figures and Tables

**Figure 1 cells-10-03551-f001:**
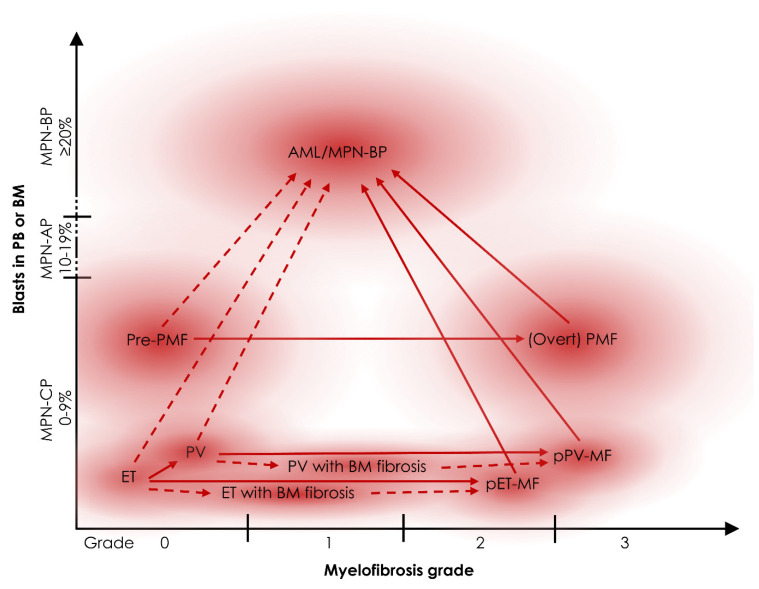
Types of MPN progression. Schematic representation of the different MPN subtypes arranged by the percentage of blasts (*y*-axis) and myelofibrosis grade (*x*-axis). The *y*-axis is subcategorized into chronic phase MPN (MPN-CP) with 0–9% blasts, accelerated phase (MPN-AP) with 10–19% blasts, and blast phase (MPN-BP) with ≥20% blasts in peripheral blood (PB) or bone marrow (BM). The types of progression are indicated in red arrows with dotted lines indicating less frequent types. AML, acute myeloid leukemia; ET, essential thrombocythemia; Pre-PMF, Prefibrotic primary myelofibrosis; PMF, primary myelofibrosis; PET-MF, post-ET myelofibrosis; PV, polycythemia vera; pPV-MF, post-PV myelofibrosis.

**Figure 2 cells-10-03551-f002:**
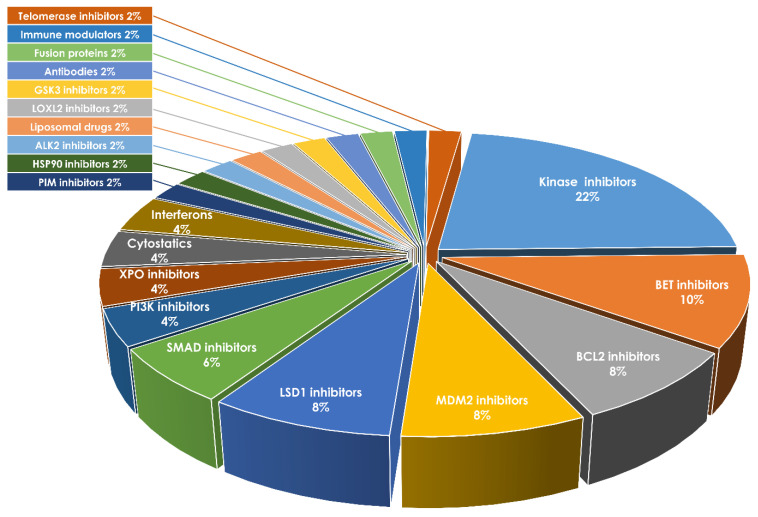
Ongoing clinical trials in MPN. Overview of selected clinical trials in MPN within the respective drug classes (including clinical trial phases and MPN subtype). The ratio of the respective drug class is given in % (relative to the total amount of studies (*n* = 49)). Data were obtained from ClinicalTrials.gov, accessed on 10 December 2021. For more information, see Table 2.

**Table 1 cells-10-03551-t001:** Overview of prognostic scoring systems for ET, PV, and PMF/SMF.

MPN subtype ⇨	ET	PV	PMF/SMF
Risk score ⇨Parameter ⇩	Conventional/ELN [61]	IPSET [62]	IPSET-thrombosis [63]	MIPSS-ET [64]	Conventional/ELN [61]	IPSS for PV [65]	MIPSS-PV [64]	IPSS [13]	DIPSS [66]	DIPSS+ [67]	MIPSS70 [14]	MIPSS70+ [14]	MIPSS70+ v2 [68]	GIPSS [32]	MYSEC (→ sMF) [69]	MTSS (→ transplant) [70]
	Age	✓	✓	✓	✓	✓	✓	✓	✓	✓	✓					✓	✓
Sex				✓												
Clinical	Leukocytes		✓		✓		✓	✓	✓	✓	✓	✓					✓
Hemoglobin/RBC								✓	✓	✓	✓	✓	✓		✓	
Blasts in peripheral blood								✓	✓	✓	✓	✓	✓		✓	
Platelets	✓									✓	✓				✓	✓
Constitutional symptoms								✓	✓	✓	✓	✓	✓		✓	
Transfusion demand										✓						
Thrombosis history	✓	✓	✓		✓	✓	✓									
Cardiovascular risk factors			✓													
Bone marrow fibrosis											✓					
HLA-mismatched unrelated donor																✓
Karnofsky performance status																✓
(Cyto-) Genetic	JAK2V617F present or MPL/CALR absent			✓	✓			✓							✓	✓	✓
Adverse/HMR mutations			✓	✓			✓				✓	✓	✓	✓		✓
Unfavorable karyotype										✓	✓	✓	✓	✓		
VHR karyotype											✓		✓	✓		
Cytogenetic risk variable													✓			
	Time of creation (from early to recent):	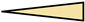	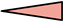	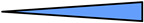	*	**

The evolution of prognostic scoring systems in ET (yellow), PV (red), and PMF/SMF (blue) characterized by the incorporated clinical and (cyto-) genetic parameters (✓). The arrows below schematically represent the time of creation from distant past (left) until present (right) and highlight the incorporation of genetic parameters over time. Abbreviations: DIPSS, dynamic IPSS; ELN, European LeukemiaNet; ET, essential thrombocythemia; GIPSS, genetically inspired prognostic scoring system; HLA, human leukocyte antigen; HMR, high molecular risk; IPSET, international prognostic score for ET; IPSS, International Prognostic Scoring System; MIPSS, mutation-enhanced international prognostic scoring system; MTSS, myelofibrosis transplant scoring system; MYSEC, myelofibrosis secondary to PV and ET prognostic model; PMF, primary myelofibrosis; PV, polycythemia vera; RBC, red blood cells; VHR, very high risk; * introduced for sMF (pPV-MF and pET-MF); ** risk score for patients undergoing transplantation.

**Table 2 cells-10-03551-t002:** Selected ongoing clinical trials in MPN.

Type	Inhibitor	MPN Subtype	Phase	NCT Number
Kinase inhibitors	Fedratinib	MF	3	NCT03755518
	Pacritinib	MF	3	NCT03165734
	Parsaclisib+Ruxolitinib	MF	3	NCT04551066
	Ruxolitinib Phosphate	MF	2	NCT01787487
	Ruxolitinib plus Enasidenib	MF	2	NCT04281498
	TL-895	MF	2	NCT04655118
	Itacitinib	MF	2	NCT04629508
	Fostamatinib	MF	2	NCT04543279
	LNK01002	MF	1	NCT04896112
	Ruxolitinib	PV	2	NCT04644211
	Ruxolitinib	ET	2	NCT04644211
	Ruxolitinib	ET/PV	2	NCT02577926
PI3K inhibitors	Parsaclisib plus Ruxolitinib	MF	3	NCT04551066
	Parsaclisib plus Ruxolitinib	MF	3	NCT04551053
BCL2 inhibitors	Navitoclax	MF	3	NCT04472598
	Navitoclax	MF	3	NCT04468984
	Navitoclax	MF	2	NCT03222609
	Palcitoclax	MF	1/2	NCT04354727
BET inhibitors	Pelabresib	MF	3	NCT04603495
	Pelabresib	MF	2	NCT02158858
	ABBV-744	MF	1	NCT04454658
	INCB057643	MF	1	NCT04279847
	INCB057643	MF	1	NCT04279847
Telomerase inhibitors	Imetelstat	MF	3	NCT04576156
SMAD inhibitors	Luspatercept	MF	3	NCT04717414
	Luspatercept	MF	3	NCT04064060
	Luspatercept	MF	2	NCT03194542
MDM2 inhibitor	Navtemadlin (KRT-232)	MF	3	NCT03662126
	Navtemadlin (KRT-232) or TL-895	MF	2	NCT04878003
	Navtemadlin (KRT-232) or TL-895	MF	1/2	NCT04640532
	Navtemadlin (KRT-232) or TL-895	MF	1/2	NCT04485260
Immune modulators	Thalidomide plus Ruxolitinib	MF	2	NCT03069326
Fusion proteins	Tagraxofusp	MF	2	NCT02268253
XPO inhibitors	Selinexor	MF	2	NCT04562870
	Selinexor	MF	1/2	NCT04562389
Antibodies	Elotuzumab	MF	2	NCT04517851
GSK3 inhibitors	9-ING-41	MF	2	NCT04218071
LOXL2 inhibitors	GB2064	MF	2	NCT04679870
Cytostatics	Decitabine	MF	2	NCT04282187
	Selumetinib/Azacitidine	MF	1	NCT03326310
Interferons	Ropeginterferon	MF	2	NCT02370329
	Ropeginterferon	ET	3	NCT04285086
Liposomal drugs	CPX-351 plus Ruxolitinib	MF	1/2	NCT03878199
ALK2 inhibitors	INCB000928	MF	1/2	NCT04455841
HSP90 inhibitors	PU-H71	MF	1	NCT03935555
PIM inhibitors	TP-3654	MF	1	NCT04176198
LSD1 inhibitors	Bomedemstat	PV	2	NCT04262141
	Bomedemstat	ET	2	NCT04254978
	Bomedemstat	ET	2	NCT04081220
	Bomedemstat	ET	2	NCT04262141

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
