# Peer review of "Progression of Myeloproliferative Neoplasms (MPN): Diagnostic and Therapeutic Perspectives"

_cells, 2021, doi:10.3390/cells10123551_

Round 1

Reviewer 1 Report

In this review paper, the authors address the question of diagnostic and treatment options for the progression of MPN. The authors are expert in the field and the review is well documented and written.

The introduction is well conducted. I suggest adding some details and updating the references:

  • In the lanes 48-49 it would be important to add the information that mutations may also occur in genes regulating the spliceosome
  • Lane 66: references should be added regarding P-selectin expression on the surface of endothelial cells. At least the references by Guadall et al., 2018 and also Guy et al., 2019 which relate experimental details on this topic.

Then authors elegantly distinguish between 2 types of MPN progression: the clinically evident progression which is presented in Figure 1 and the subclinical evolution of MPN which focuses mainly on the pre-MPN trajectory of mutated cells.

In the following sections some references should be adjusted. Mainly it is not adapted in a review paper to cite other reviews.

  • Lane 219: instead of the reference 42 which is a review it should be more adapted to cite the original works reporting the correlation of IL2R and IL8 with survival
  • Lane 221: same comment for reference 43. Please provide the references of original papers.
  • Lane 361: the reference 84 is an opinion paper that should be removed. It does not relate to LSD1. Whatever, prefer to cite the original work.
  • Lanes 349: the reference 77 should be removed and changed for the references by Kiladjian et al., 2008 and Quintas-Cardama et al., 2013 which were the first to report molecular responses under IFNa therapy.

Of note, the authors, in an interesting paragraph, discuss on innovative non-invasive approaches to evaluate bone marrow fibrosis using PET-CT and others.

Several formal modifications:

  • In lane 136 the abbreviated term AB should be explained somewhere.
  • There are several references which do not appear correctly as in lanes 115, 259 and 365
  • Table 1 should be modified as it is impossible to read the 2nd row correctly
  • Table 2 may be removed as it does not provide important information for the reader.

Reviewer 2 Report

The review by Baumeister et al. entitled "Progression of myeloproliferative neoplasms (MPN) : Diagnostic and therapeutic approaches" is well written, clear, interesting, quite complete and thus deserves publication in Cancers, provided the authors make a number of revisions.

The main changes concern the discussion of inflammation, and the main cytokines involved. Several authors have extensively covered this important subject, who are either not cited in the review, or not sufficiently. For instance, a special issue has been dedicated to inflammation in MPN, that clearly explained that cytokines played an essential role in clonal progression in MPN (likely more important than mutations). Some of these  papers (see below) should be discussed and cited.

Moreover, a number of cytokines have been demonstrated to be of critical importance for clonal progression in MPNs ; those include but are not limited to IL-8, IL-1beta and TNFalpha (mentioned by the authors, page 5) (examples : IL-15, HGF, GRO-alpha, EGF, cf. papers cited below). In terms of therapy, it is important to recognize that most of these cytokines are insensitive to JAK inhibitors. In contrast, IFN-α counters many of these cytokines (via the repression of IL-1β, notably), which likely contributes to the efficacy of Ropeginterferon. Similarly, drugs exist that inhibit IL-1b, for instance (some cited in the review) and this approach should also be discussed.

In fact, knowing which of the main clone progression-associated cytokines are produced excessively by a given patient and targeting these cytokines as well as the patient’s driving mutation(s) may be the best perspective for curative MPN treatments. The authors may want to discuss this better.

Minor points :

  • Using "AB" for "allelic burden" is not usual and not necessary ("allelic burden" is not used frequenly in the manuscript). Please write allelic burden in all letters.
  • In several instances there are mistakes in references in the text, that appear as "Error ! Reference not found" : cf. page 3 line 115 ; page 6 line 259 ; page 9 line 365. This should not be corrected.
  • Page 4 line 151 : "The driver mutation AB" is not proper English. Please correct.
  • Page 4 line 155 : "… the AB was well below the detection of other methods such as Sanger sequencing" : this part of the sentence should be deleted, since there is international consensus that Sanger sequencing should no longer be used for the detection of mutations in MPN. NGS or ddPCR are the preferred methods (as stated by the authors, line 151).
  • Table 1 page 7 : the heads of the table (risk scores for ET, PV, PMF) are not readable. Please correct.

Examples of papers omitted

Mediators of Inflammation in Myeloproliferative Neoplasms: State of the Art.

Mediators Inflamm. 2015:964613. doi: 10.1155/2015/964613. (Special Issue)

https://www.hindawi.com/journals/mi/si/329170/

Pathogenesis of Myeloproliferative Neoplasms: Role and Mechanisms of Chronic Inflammation. Mediators Inflamm. 2015;2015:145293. doi: 10.1155/2015/145293.

MPNs as Inflammatory Diseases: The Evidence, Consequences, and Perspectives. Mediators Inflamm. 2015;2015:102476. doi: 10.1155/2015/102476.

Anti-Glucosylsphingosine Autoimmunity, JAK2V617F-Dependent Interleukin-1β and JAK2V617F-Independent Cytokines in Myeloproliferative Neoplasms. S Allain-Maillet et al. Cancers (Basel). 2020;12(9):2446. doi: 10.3390/cancers12092446.

Longitudinal Cytokine Profiling Identifies GRO-α and EGF as Potential Biomarkers of Disease Progression in Essential Thrombocythemia. Øbro NF et al. Hemasphere. 2020;4(3):e371. doi:10.1097/HS9.0000000000000371.

Round 2

Reviewer 2 Report

The paper has been revised appropriateley and can now be published.